



# A structural self-organizing map (S-SOM) algorithm for weather typing

Quang-Van Doan[1] Hiroyuki Kusaka[1], Takuto Sato[2], Fei Chen[3]

[1] Center for Computational Sciences, University of Tsukuba, Tsukuba, Ibaraki, Japan.
[2] Graduate School of Life and Environmental Sciences, University of Tsukuba
[3] Research Applications Laboratory, National Center for Atmospheric Research, Boulder, Colorado, USA.

*Correspondence to*: Quang-Van Doan (doan.van.gb@u.tsukuba.ac.jp)

**Abstract.** In this study, we propose a novel structural self-organizing map (S-SOM) algorithm for synoptic weather typing. A novel feature of the S-SOM compared with traditional SOMs is its ability to deal with input data that have spatial or temporal

structures. In detail, the search scheme for the best matching unit (BMU) in a S-SOM is built based upon a structural similarity (S-SIM) index rather than by using the traditional Euclidean distance (ED). S-SIM enables the BMU search to consider the correlation in space between weather states, such as the location of highs of lows, that is impossible when using ED. The S-SOM performance is evaluated by multiple demo simulations of clustering weather patterns over Japan using the ERA-Interim sea-level pressure data. The results show the superiority of the S-SOM compared with a standard SOM with ED (or ED-SOM)

in two respects: clustering quality based on silhouette analysis and topological preservation based on topological error analysis. The superior performance of the S-SOM compared with an ED-SOM is probably independent of both the input data and SOM configuration.

## 1. Introduction

In recent years, there has been an increasing number of self-organizing map (SOM) applications in climatology studies. The

SOM was initially developed by Kohonen (1982) as an unsupervised data mining method. SOMs are used to discover patterns intrinsic to input data by projecting them into a map (usually two dimensional), and the nodes on the map represent the most important features of the input space. One standard application of SOMs in climatology is "objective" synoptic weather typing (Sheridan and Lee, 2011). Here synoptic circulation data, typically sea-level pressure (SLP) or geopotential height, are used to generate a small enough number of representative weather states that can be readily handled by sequential analysis.

SOMs are used for diverse purposes, from discovering the links between synoptic circulation and climatic variability to statistical dynamical downscaling, climate prediction, and weather forecasting. For example, with an SOM, Horton et al. (2015) found that changes in the frequency of geopotential height patterns since the 1980s have modified extreme temperature trends



in some northern hemisphere regions. Also, SOMs have been used to discover the association between changes in rainfall and shifts in large-scale circulation patterns (e.g., Alexander et al., 2010; Lennard and Hegerl (2015), Swales et al., 2016; Nguyen-Le et al., 2019; Luong et al., 2020). In addition, SOMs are also used as a statistical downscaling method for the future climate by associating the changes in the frequency of synoptic occurrences with surface variables (e.g., Gibson et al. 2016, Ohba et al., 2016; 2019). Borah et al. (2013) developed a probabilistic prediction scheme for the Indian summer monsoon intraseasonal oscillation using an SOM-based technique. Chang et al. (2014) used a hybrid SOM and dynamic neural networks for nowcasting rainfall in Taiwan. Nguyen-Le et al. (2017) used a hybrid system of numerical weather prediction (NWP) and an SOM to forecast heavy rainfall for up to a week for Kyushu, Japan. A hybrid NWP and an SOM were also used by Ohba et al. (2018) as a system for the medium-range forecasting of wind ramps in Japan. Lagerquist et al. (2017) proposed, for the first time, a way to obtain a real-time extreme wildfire weather forecast in the US using an SOM technique.

The SOM algorithm consists of repeatedly learning processes that gradually update the nodes in the output map until they converge to a stable solution, which is also, as expected, the "best" representative of the input space. At each learning step, the SOM selects an input vector, usually randomly, and then searches for a node in an SOM map that best matches that vector. In this task, nodes in the output map compete with each other to find the node most "similar" to the input vector. The "winning" node is then called the best matching unit (BMU). Next, training is implemented by making the BMU and its neighbors closer to the input vector. The training task is governed by the learning rate and neighborhood function. Needless to say, searching the BMU is a crucial part of the SOM algorithm as it affects the sequential training process and the quality of the final SOM outcome.

Traditional SOMs use the Euclidean distance (ED) to search for the BMU, where the "closest" node to an input vector in terms of ED will be assigned as the BMU. This method is simple and computationally effective. Moreover, ED is very popular and widely used as a quantitative similarity metric when comparing two objects. ED is commonly used in many machine learning algorithms. It is also the basis of many clustering algorithms such as K-means, affinity propagation. Nevertheless, ED has severe shortcomings when it is used to compare "structured" signals, i.e., those with spatial or temporal orders, such as time series and 2-dimensional images. Despite these shortcomings, ED has been influential and widely used. A reason for its popularity is that the prevailing attitudes towards ED seem to range from "it's easy to use and not so bad" to "everyone else uses it." (Wang and Bovik, 2009)

This weakness of ED becomes crucial in climatology studies where most of the data are spatially and temporally structured, e.g., weather maps, and time series. Intuitively, a similarity measure based on ED might lead to the degradation of the spatial correlations between air pressure patterns, such as the location of highs or lows (Fig. A1). Thus, a BMU search scheme using ED might result in a mis-determination of the "winning node", which would critically affect the performance of SOMs. Several



alternative versions of the SOM have been developed since Kohonen (1984). These include the generative topographic map (Kaski et al., 1997) and the time adaptive self-organizing map (Shah-Hosseini et al., 2003; 2011). However, such SOM versions have focused on parameterization schemes such as learning rate or neighborhood functions in the training process. No studies have addressed the fundamental issue of the ED in a BMU search.

Therefore, this study proposes a novel SOM algorithm, which we call a structural SOM (S-SOM). The advantage of S-SOM compared with traditional SOMs is that an S-SOM is able to deal with "structural" input data, i.e., data with spatial or temporal relationships. To accomplish this, the S-SOM incorporates a BMU search scheme that is implemented based on a structural similarity index rather than on the traditional ED.

The structural similarity (S-SIM) index, which was first introduced by Wang et al. (2004), and which is being increasingly used in the signal processing field, has an advantage over ED in its ability to detect structural correlations in the pair data. We set up multiple test simulations with different SOM configurations to evaluate the performance of the S-SOM by comparison with the traditional ED-SOM when classifying sea-level pressure patterns over the Japan region. Quantified metrics such as silhouette analysis and topological errors are used to assess SOM performance. The remainder of this paper is structured as follows. Section 2 describes the novel S-SOM algorithm; section 3 presents the test simulation configuration and evaluation metrics. Results are presented and discussed in section 4. Concluding remarks are provided in section 5.

## 2. Structural SOM algorithm

Our proposed S-SOM algorithm is shown in Figure 1. It follows the procedure initially proposed by Kohonen (1982) and used in many application studies. An S-SOM starts with the configuration and initialization of SOM nodes, and establishing the number of training iterations. The training consists of three main steps: selecting an input vector, finding the best matching unit for the input vector, then updating the weight vectors of SOM nodes by using parameters, i.e., learning rate and neighborhood function. The learning rate is a real number and decreases as the number of iteration steps increases. The difference between S-SOM and traditional SOM implementation is that we propose a new scheme for finding a BMU. In this scheme, we use a similarity index that can deal with structural data such as two-dimensional air pressure distribution instead of using ED to compare the similarity between vectors.

The new BMU search scheme is based on competition among SOM nodes so that a node with the highest structural similarity (S-SIM) index in relation to an input vector will be assigned as the BMU. The S-SIM was first introduced by Wang et al. (2004) as a method for predicting the perceived quality of digital television and cinematic pictures. The basic model was developed in the Laboratory for Image and Video Engineering at the University of Texas at Austin and further developed





jointly with the laboratory for Computational Vision at New York University. The S-SIM index is designed to improve on traditional methods such as the peak signal-to-noise ratio and mean squared error, i.e., methods based on ED, to detect similarities in "structural" signals such as images. The S-SIM formula is based on three comparison measurements between two vectors $x, y$, luminance ($l$), contrast ($c$), and structure ($s$).

$$SSIM(x,y) = [l(x,y)^{\alpha} \times c(x,y)^{\beta} \times s(x,y)^{\gamma}] \tag{1}$$

90  Here individual comparison functions are:

$$l(x,y) = \frac{2\mu_x\mu_y + c_1}{\mu_x^2 + \mu_y^2 + c_1} \tag{2}$$

$$c(x,y) = \frac{2\sigma_x\sigma_y + c_2}{\sigma_x^2 + \sigma_y^2 + c_2} \tag{3}$$

$$s(x,y) = \frac{\sigma_{xy} + c_3}{\sigma_x\sigma_y + c_3} \tag{4}$$

Here, $\mu_x, \mu_y$ are the average, and $\sigma, \sigma_y$ are the variance of vectors $x, y$, respectively; $c_1, c_2, c_3$ are parameters to stabilize division with a weak denominator. To simplify the model, here we set $c_1 = c_2 = c_3 = 0$. and weights $\alpha = \beta = \gamma = 1$ to reduce the original formula to:

$$SSIM(x,y) = \frac{(2\mu_x\mu_y)(\sigma_{xy})}{(\mu_x^2 + \mu_y^2)(\sigma_x^2 + \sigma_y^2)} \tag{5}$$

From the definition, S-SIM ranges from -1 to 1, where 1 indicates perfectly similar, and vice versa. The S-SIM has been
95  repeatedly shown to outperform ED significantly in terms of accuracy. Wang and Bovik (2009) pointed out that an S-SIM provides powerful, easy-to-use, and easy-to-understand alternatives to traditional ED for dealing with specific kinds of data that are spatially and temporally ordered. Recently, S-SIMs are attracting attention as a "new-generation" similarity metric in hydrological and meteorological studies (e.g., Mo et al., 2014; Han and Szunyogh, 2018)



## 3. Model configuration and quality measurements

### 2.1 Data and experiment settings

ERA-Interim (Dee et al., 2011) reanalysis daily mean-sea-level pressure (MSLP) data for 1979–2019 over the Japan region (latitude 20 to 50˚N and longitude 115°E to 165°E; see Figure 2) are used for demo simulations of S-SOM. The original MSLP data at a 0.75° resolution on a regular grid were interpolated to an Equal-Area Scalable Earth-type grid at a spatial resolution of 100 km. This interpolation method has been commonly applied in high-latitude regions (Lynch et al., 2016; Gibson et al., 2018). The data are divided according to the four seasons: winter (December, January, February or DJF), spring (March, April, May or MAM), summer (June, July, August, or JJA), autumn (September, October, November, or SON).

The SOM grid topology consists of one-dimensional nodes. The training was carried out with 5000 iterations and with the learning rate start-point at 0.01 (decreased exponentially to 0). The Gaussian function is used as the neighborhood function. A random initialization scheme was used. For completeness, we also train our SOMs with various configurations of n-nodes = 4, 5, …, 20.  With four seasons, and 17 (node configurations) and 2 (S-SOM and ED-SOM), we conduct a total of 4 x 17 x 2 = 136 of SOM runs.

### 2.2 Quality evaluation

We evaluate the performance of an S-SOM versus an ED-SOM, focusing on two different aspects. One is the capability as a clustering method, which we investigate by using silhouette analysis; the other is the ability to preserve the topology of input space, which we investigate by analyzing topographical error. We select the evaluation metric based on its widespread use in clustering evaluation (silhouette analysis) and the SOM characteristics.

Silhouette refers to a method for the interpretation and validation of consistency within clusters of data. The technique provides a succinct graphical representation of how well each object has been classified (Rousseeuw, 1987). The silhouette value is a measure of how similar an object is to its cluster (cohesion) compared with other clusters (separation). The silhouette coefficient is calculated using the mean intra-cluster distance (a) and the mean nearest-cluster distance (b) for each sample. The silhouette coefficient for a given sample is then defined as $s = (b - a)/(\max(a, b))$. The value ranges from −1 to +1, where a high value indicates that the object is well matched to its cluster and poorly matched to neighboring clusters. If most objects have a high value, then the clustering configuration is appropriate. If many points have a low or negative value, then the clustering configuration may have too many or too few clusters. Silhouette coefficients near +1 indicate that the sample is far from the neighboring clusters. A value of 0 indicates that the sample is on or very close to the decision boundary between two neighboring clusters, and negative values indicate that those samples might have been assigned to the wrong cluster.





One important goal of the SOM algorithm is to preserve the topological features of the input space in the output space. The topological error (TE) is defined as the average geometric distance between the winning and the second-best matching nodes in the SOM (Gibson et al., 2018). If the nodes are next to each other, we say that the topology has been preserved for this input; otherwise, it is counted as an error. The total number of errors divided by the total number of inputs gives the number of topographic errors of the map. TE measures how well the structure of the input space is modeled by the SOM. Primarily, it evaluates the local discontinuities in the mapping, i.e., $TE = 1/n \sum_{i=1}^{n} d_i$ . Here, $d_i$ is the distance between the best matching and second matching units to an input vector $x_i$; $n$ is the total number of input vectors. The best value of TE is 1 meaning that BMU and SMU are neighbors. A larger TE means a higher topographical error. A lower TE indicates a more topologically ordered (i.e., more "self-organized") SOM plane.

## 4. Results

Before analyzing the SOM results, we examine the performance of similarity indices, i.e., S-SIM and negative ED (since ED is a measure of distance, negative ED is a measure of similarity), in terms of how they detect the difference between SLP maps. The similarity index for each pair of SLP data is calculated and normalized from 0 – 1, where 1 means exactly the same, and 0 means most different (indicating the value for the pair that are furthest from each other). Histograms of the distribution of values, which we call normalized "distinguishing" distributions (NDD), are shown in Figure 3 for four datasets, i.e., DJF, MAM, JJA, and SON. For each dataset, the NDDs for ED and S-SIM are compared with each other.

The comparison provides two major insights regarding the performance of the S-SIM and ED in terms of distinguishing differences between maps. First, the NDDs of the S-SIM appear to spread over two tails, whereas those of ED tend to concentrate around their averages. The standard deviations of four S-SIM NDDs are about 0.17 – 0.18, which is consistently higher than the ED values of 0.09 – 0.10 (Table 1). This means that with an S-SIM, we can obtain a higher range of similarity values for pairs of data, which indicates a greater ability to distinguish one from others. This ability is consistent among DJF, MAM, JJA, and SON, implying that the higher discrimination ability of the S-SIM is likely data-independent.

Second, the shapes of S-SIM NDDs appear to vary, whereas those of ED look identical for DJF, MAM, JJA, and SON. In other words, an S-SIM is able to effectively recognize seasonal variability, but ED does not. The NDDs of ED have the same mean values at about 0.69 to 0.71, and skewness at about -0.65 to -0.79 among seasons. Meanwhile, S-SIM NDDs vary in terms of mean, ranging from 0.53 to 0.70, and skewness, ranging from -0.09 to -0.73. In particular, the S-SIM skewness of DFJ and JJA is lower than that of MAM and SON. In MAM and SON, the skewness is close to 0, meaning the NDDs are almost symmetric. The NDDs of an S-SIM are more reasonable in the sense of physical interpretation. In DJF, the weather of Japan is dominated by the winter-type air pressure (high in the west and low in the east), with a small number of exceptions.





This explains why the daily SLP in DJF looks similar most of the time; the mean of the DJF NDDs is higher than in other seasons; the skewness is highly negative. The same weather trend is observed in the summer months (JJA). Meanwhile, in MAM and SON, which are transition periods between winter and summer and vice versa, the weather variability is higher, and there are no dominant patterns during these times. This reality is reflected in the S-SIM asymmetric NDDs in MAM and

SON.

**Table 1. Statistical indices of normalized discrimination distributions**

|  | DJF | | MAM | | JJA | | SON | |
|---|---|---|---|---|---|---|---|---|
|  | S-SIM | ED | S-SIM | ED | S-SIM | ED | S-SIM | ED |
| **MEAN** | 0.70 | 0.69 | 0.53 | 0.69 | 0.61 | 0.68 | 0.55 | 0.71 |
| **STANDARD DEVIATION** | 0.17 | 0.10 | 0.18 | 0.10 | 0.16 | 0.09 | 0.17 | 0.09 |
| **SKEWNESS** | -0.73 | -0.78 | -0.09 | -0.65 | -0.42 | -0.71 | -0.13 | -0.79 |
| **KURTOSIS** | 0.04 | 0.75 | -0.70 | 0.47 | -0.31 | 0.70 | -0.60 | 0.87 |

Next, we analyze the outcome of SOMs to ensure that the results are physically reasonable and match the common perception about seasonal weather patterns in Japan. Figure 4 shows the SLP patterns typed by SOMs (node number =4) for DJF. At first glance, S-SOM is generally able to detect the dominant weather pattern in winter. It is well known that the Japanese winter is characterized by the Siberian High, which develops over the Eurasian Continent, and the Aleutian Low, which develops over the northern North Pacific. Prevailing northwesterly winds cause the advection of cold air from Siberia, bringing heavy

snowfall to western Japan and sunny weather to the eastern side (www.data.jma.go.jp/gmd/cpd/longfcst/en/tourist_japan.html). Such a dominant pattern is likely to be well detected by S-SOM (Fig. 4a) and ED-SOM (Fig. 4f).



An interesting difference between SOMs is that S-SOM appears to estimate a more "ordered" clustering of nodes (Fig. 4e), characterized by a dominant node (N1) accompanied by non-dominant ones. Meanwhile, for ED-SOM, the size of clusters is relatively identical, underlining the presence of more "flat" clustering (Fig. 4j). This trend is consistent seen in other SOM

runs (i.e., node numbers greater than 4). "Ordered" clustering by S-SOM and "flat" clustering by ED-SOM are also recognized for JJA but to a lesser extent (see Fig A3). The physical explanation for this is that the Japanese summer is characterized by two sub-periods. Early sub-period is rainy caused by a stationary front, called the Baiu front, where a warm maritime tropical air mass meets a cold polar maritime air mass. In the second sub-period, the North Pacific High extends northwestward around Japan, bringing hot and sunny conditions to the country, and the number of tropical cyclones passing the country peaks in

August. Different to DJF and JJA, for MAM, and SON, which are transition seasons, the difference, i.e., "ordered" and "flat" clustering, is not apparent for S-SOM and ED-SOM.

Moreover, silhouette analysis shows that S-SOM performs consistently better than ED-SOM in clustering SLP patterns. It should be remembered that a silhouette analysis quantifies the quality of clustering by calculating how properly an object is assigned to a cluster but not to the others. Figure 5 reveals two crucial points: one is the thickness of clusters (y-axis), and the

other is the silhouette coefficient value. As explained above, for DJF, S-SOM tends to estimate one dominant winter-type SLP pattern combined with minor exception patterns. This is different from ED-SOM, which tends to predict more "flat" clusters with the same thickness. From the silhouette plot, S-SOM clustering makes more sense than ED-SOM. The dominant Japanese winter-type pattern can be easily identified by looking at the S-SOM plot but this is not the case with the ED-SOM plot.

An important point here is that despite the "large" cluster, the silhouette values of S-SOM tend be to consistently higher than

those of ED-SOM (Fig. 5). This is highly counter-intuitive because generally, if the cluster is large, there is a higher possibility of data points being assigned to the wrong cluster. Also, Figure 6 summarizes the silhouette score, compares both SOMs and highlights two critical points: 1) S-SOM is consistently superior to ED-SOM for all seasons and all SOM node configurations, demonstrating that S-SOM offers higher quality clustering than ED-SOM, which is consistent and independent of simulations; 2) although not shown, the S-SOM score for DJF and JJA months are higher than for MAM and SON, whereas those of ED-

SOM are not seasonally different. We speculate that, for DJF or JJA, data are more ordered with a dominant pattern, and S-SOM performs better when classifying such patterns than for transition seasons like MAM and SON.

As a measure of the topology preservation of SOM, Topographic error (TE) indicates the lower error (higher topological preservation) of S-SOM compared with ED-SOM (Fig. 7). Unlike with the silhouette score (Fig. 6), the difference between the TE values with S-SOM and ED-SOM is less noticeable and less consistent among SOM-node configurations and input

data. A lower TE is always seen for MAM and JJA with almost all SOM size settings. However, for DJF, S-SOM has a higher TE especially when it has a small size of 4 or 5. For SON, there is no apparent difference between S-SOM and ED-SOM.





Overall, we do not see a consistent trend in TE between S-SOM and ED-SOM. As TE is known to strongly depend on the neighborhood function (Gibson et al., 2018), the similarity detection scheme might have less impact. Nevertheless, as shown here, although it is still uncertain, S-SOM exhibits a lower TE in most cases. We can say that S-SOM can provide comparable

if not better topology preservation. We also suggest that future studies are needed to clarify the topology preservation ability of S-SOM and its dependence on the similarity index.

## 5. Conclusions and remarks

In this study, we developed a novel SOM algorithm (S-SOM) for synoptic weather typing. An advantage of S-SOM compared with traditional SOMs is that S-SOM uses a BMU search scheme built based on the structural similarity index (S-SIM) instead

of the traditional Euclidean distance (ED). To evaluate the performance of S-SOM, we conducted multiple demo simulations to cluster weather patterns over the Japan region using ERA-Interim SLP data. The results obtained with S-SOM are compared with those obtained with the traditional SOM using the ED, i.e., ED-SOM.

The results demonstrated the superiority of S-SOM compared with ED-SOM in two respects: clustering quality and topological preservation. Clustering quality is quantitively assessed by a silhouette analysis and the results show consistently higher

silhouette scores for S-SOM than with ED-SOM. In the same way, there appear to be fewer topographic errors with S-SOM than with ED-SOM, implying the superior ability of S-SOM as regards topological preservation of the input space. Moreover, this superior performance of S-SOM is consistent for all simulations and SOM map configurations.

From these results, we highlight the effectiveness of using S-SOM rather than traditional SOM, at least when input data are featured by spatial distributions. Although this study did not assess the performance of S-SOM on time series, we believe that

S-SOM can also be effective for temporally distributed data. The S-SOM performance with time series should be assessed in a further study. Moreover, although S-SOM has been developed primarily for climatology studies, it can also be used in other fields. We expect it will constitute the new standard SOM when dealing with "structural" input data.

### Code availability

The exact version of the model used to produce the results used in this paper is archived on Zenodo

(https://zenodo.org/deposit/4039883#), as are input data and scripts to run the model and produce the plots for all the simulations presented in this paper.



**Author contribution**

Quang-Van Doan designed the model and developed the model code. Hiroyuki Kusaka and Takuto Sato helped to design, carry out the test experiment and analyze the results. Fei Chen helped to analyze the results. Quang-Van Doan prepared the

manuscript with contributions from all co-authors.

**Acknowledgments**

This research was funded by the Environment Research and Technology Development Fund JPMEERF20192005 of the Environmental Restoration and Conservation Agency of Japan. Quang-Van Doan is grateful for support from JSPS KAKENHI Grant Number JP20K13258 and JSPS KAKENHI Grant Number JP19H01155. Fei Chen is grateful for support from the Water

System Program at the National Center for Atmospheric Research (NCAR), NASA IDS Grant#80NSSC20K1262, USDA NIFA Grants 2015-67003-23508 and 2015-67003-23460, NOAA NA18OAR4590398, and NSF Grant #1739705.

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

```

**Fig. 1. The S-SOM algorithm.**



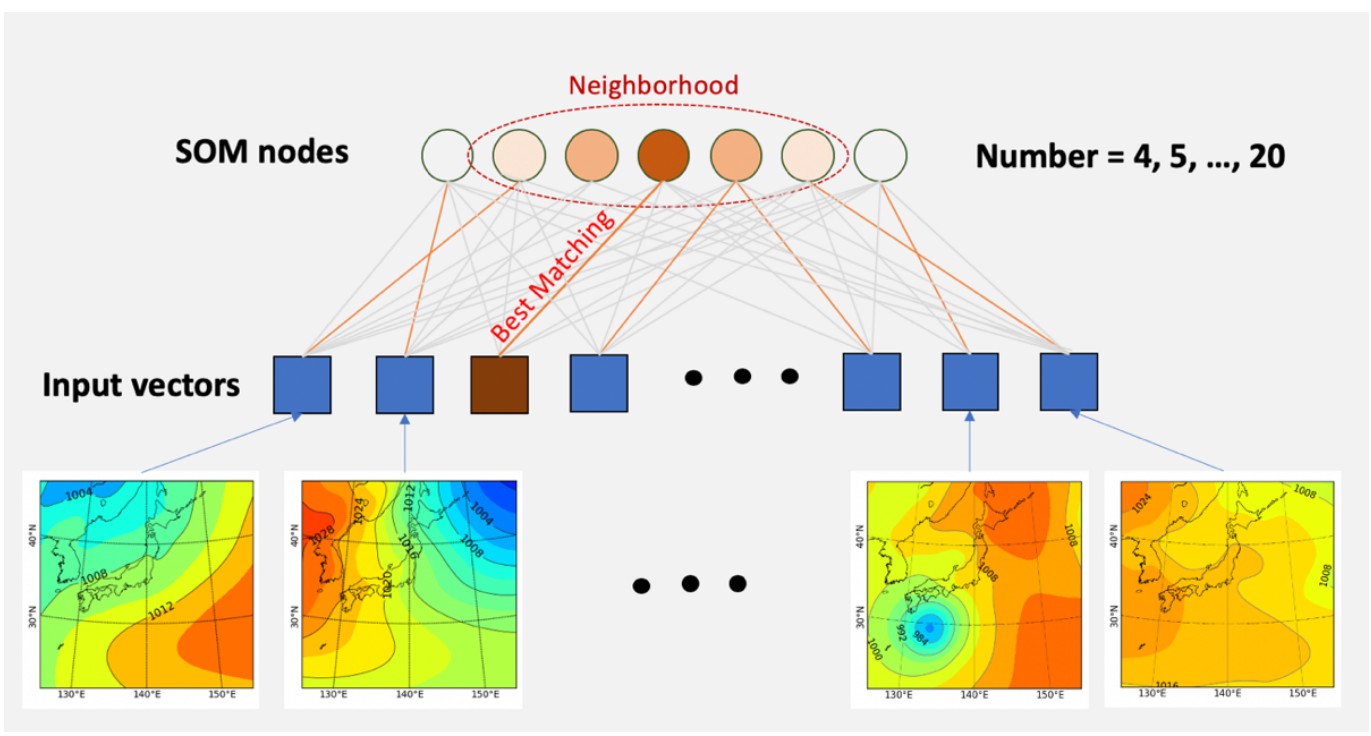


**Fig. 2. Illustration of SOM configuration and simulation settings. Used data is daily (at 00 UTC) ERA Interim sea level pressure (from 01 Jan 1979 to 01 December 2019) divided into four seasons: winter (December-January-February), spring (March-April-May), summer (June-July-August), and autumn (September-October-November)**




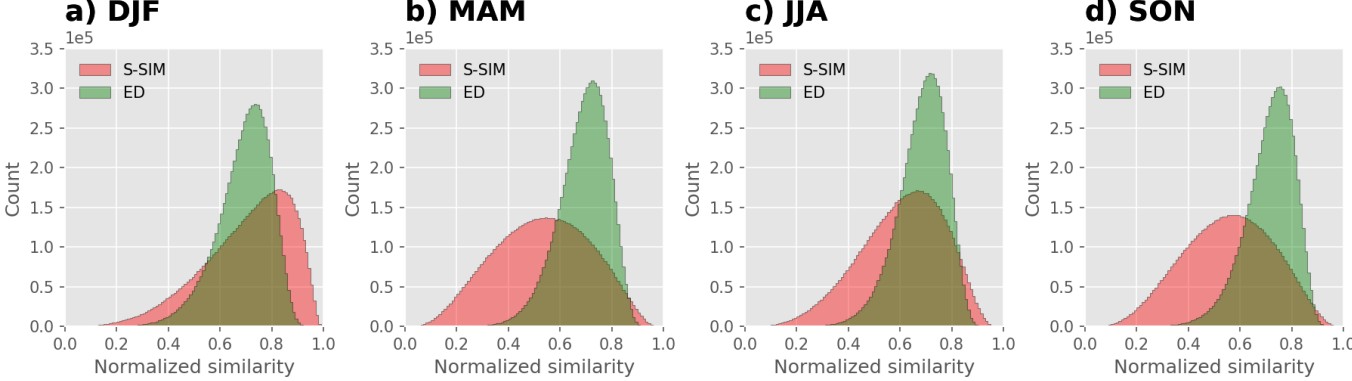

**Fig. 3. Comparison of normalized distinguishing distributions (NDDs) of S-SiM and ED for four-season data. An NDD is the distribution of similarity values indicating the degree to which data points in a data population are similar (or separated) from each other. With a population size of N, NDD has (N-1)! Values, because both S-SIM and ED are symmetric measures and not including self-similarity. Values are normalized from 0 to 1, i.e., $s_i = (s_i - min\{s\})/(max\{s\} - min\{s\})$, with $i = 1, 2, .., N$. The maximum similarity is 1, i.e., perfectly similar, and the minimum similarity is 0, i.e., the lowest similarity between a pair of data points. Thus, the minimum similarity is dependent on the measure (S-SOM or ED) and data (DJF, MAM, JJA, or SON).**




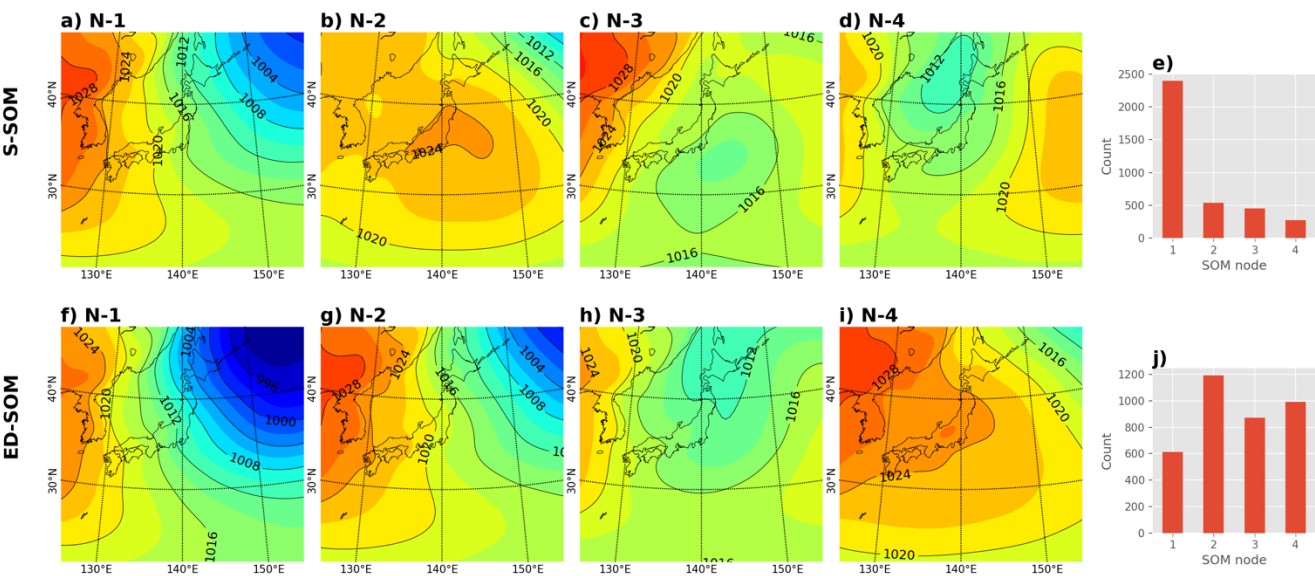

**Fig. 4. Winter (DJF) MSLP pattern revealed by SOMs. Upper plots, from a) to d), show S-SOM patterns (with 4 nodes); e) shows the number of daily MSLPs classified as nodes 1 to 4. The lower plots show the same information for ED-SOM.**



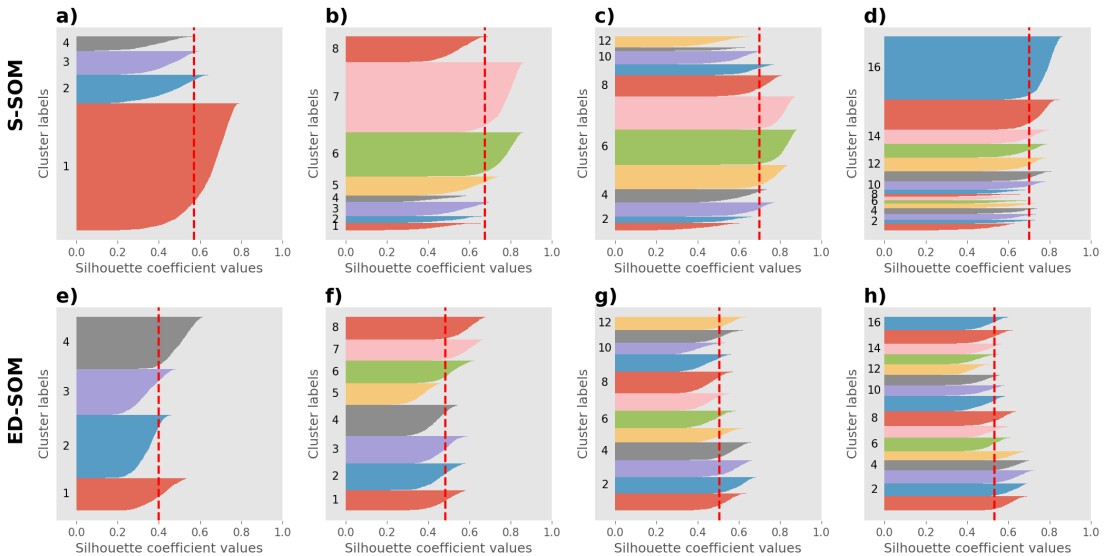

**Fig. 5. Silhouette plot for S-SOM and ED-SOM clustering for winter month DJF. a) to d) show results from simulations with SOM node numbers of 4, 8, 12, 16, respectively. e) to h) show the corresponding results for ED-SOM. The vertical red dashed lines in each plot indicate the silhouette score.**




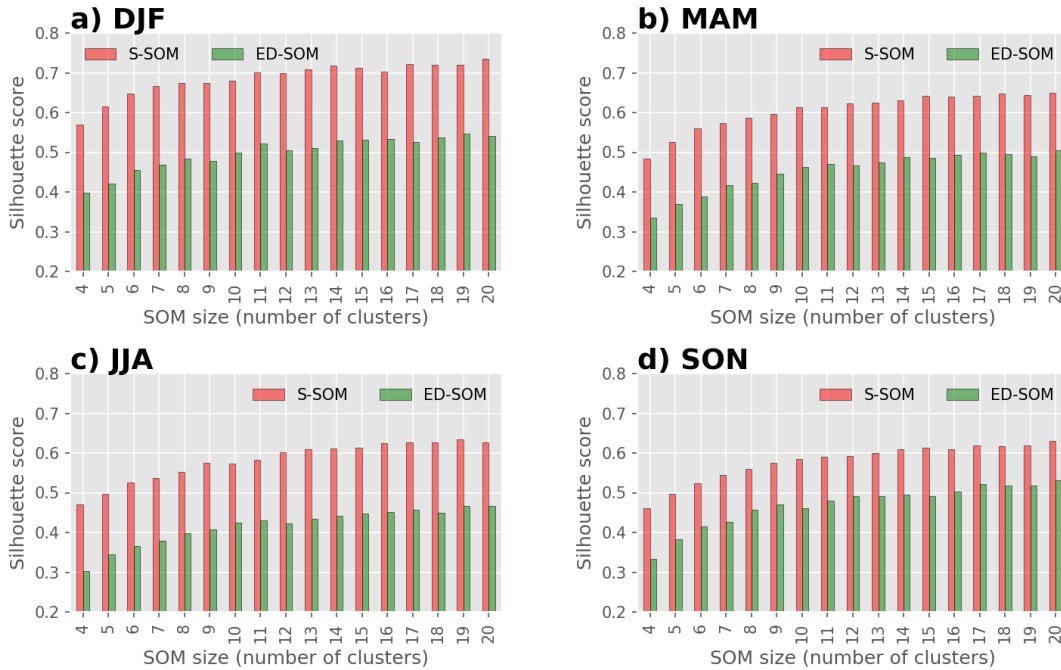

**Fig. 6. Comparison of silhouette scores of S-SOM and ED-SOM for all SOM size configurations and four seasons, i.e., DJF, MAM, JJA, SON. In each plot, the x-axis indicates a different SOM simulation (size configuration), and the y-axis indicates silhouette score values.**





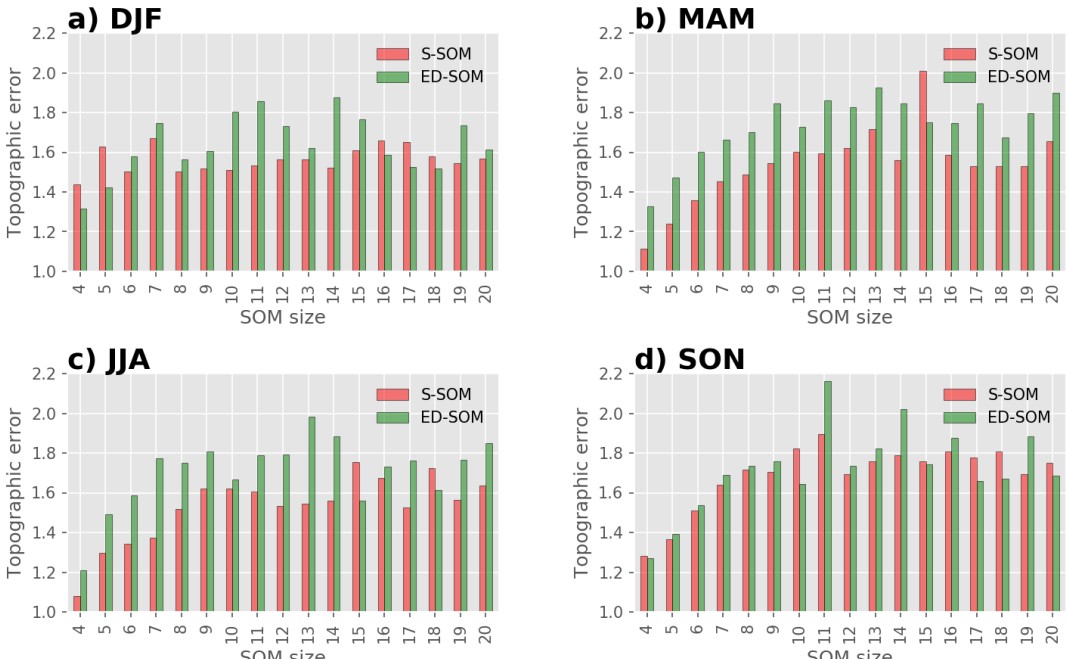

**Fig. 7. Comparison of topographic errors of S-SOM and ED-SOM for all SOM size configurations and four seasons, i.e., DJF, MAM, JJA, SON. In each plot, the x-axis indicates a different SOM simulation (size configuration), and the y-axis indicates topographic errors.**






**Appendices**

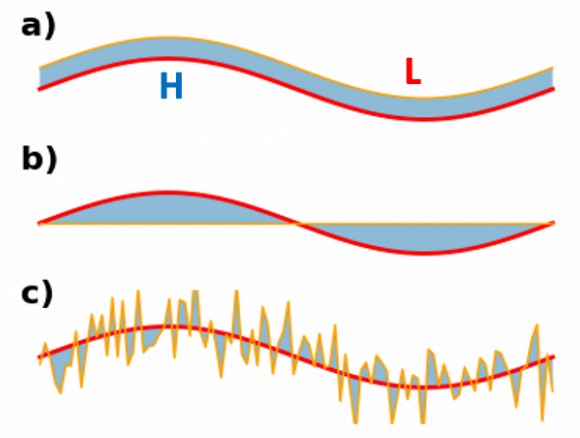


**Fig. A1. Example faults with Euclidean distance (ED) when distinguishing data that have a spatial and temporal correlation. Suppose we have two distributions represented by red and orange lines in each plot a) b) and c), where H and L are the location of a High and a Low, respectively. In a) b) and c) the red and orange lines have the same ED; meanwhile, if use the Structural Similarity (S-SIM) index to compare the lines, one will have the highest S-SIM value**
**in a), followed by c); the S-SIM values in both a) and c) are much higher than that in b).**





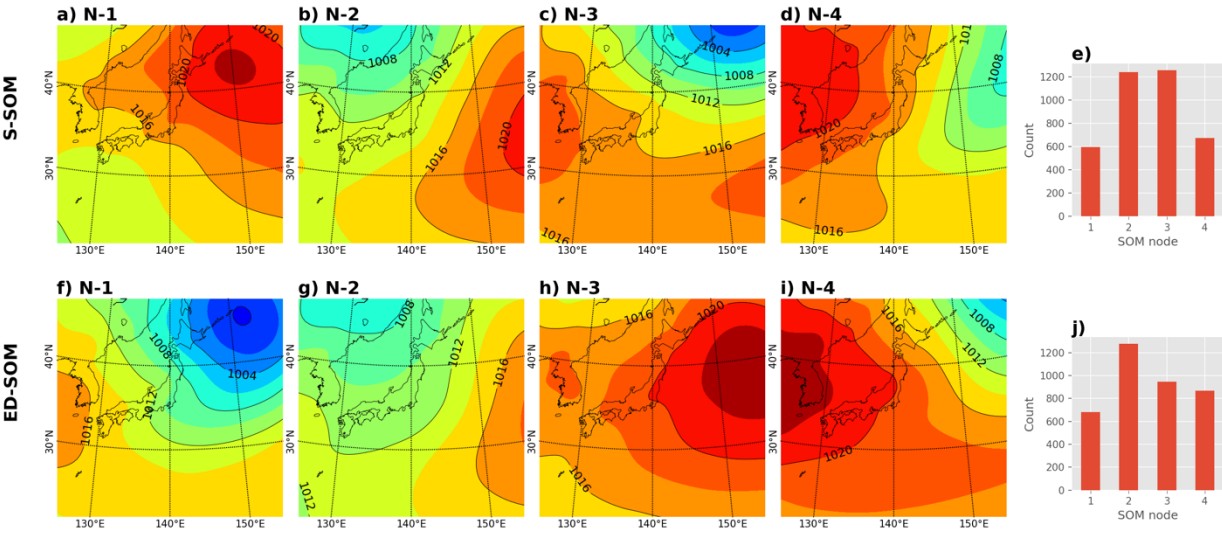


**Fig. A2. Spring (MAM) MSLP pattern revealed by SOMs. Upper plots, from a) to d), show S-SOM patterns (4 nodes); e) Shows the number of daily MSLPs classified as nodes 1 to 4. The lower plots show the corresponding number for ED-SOM.**


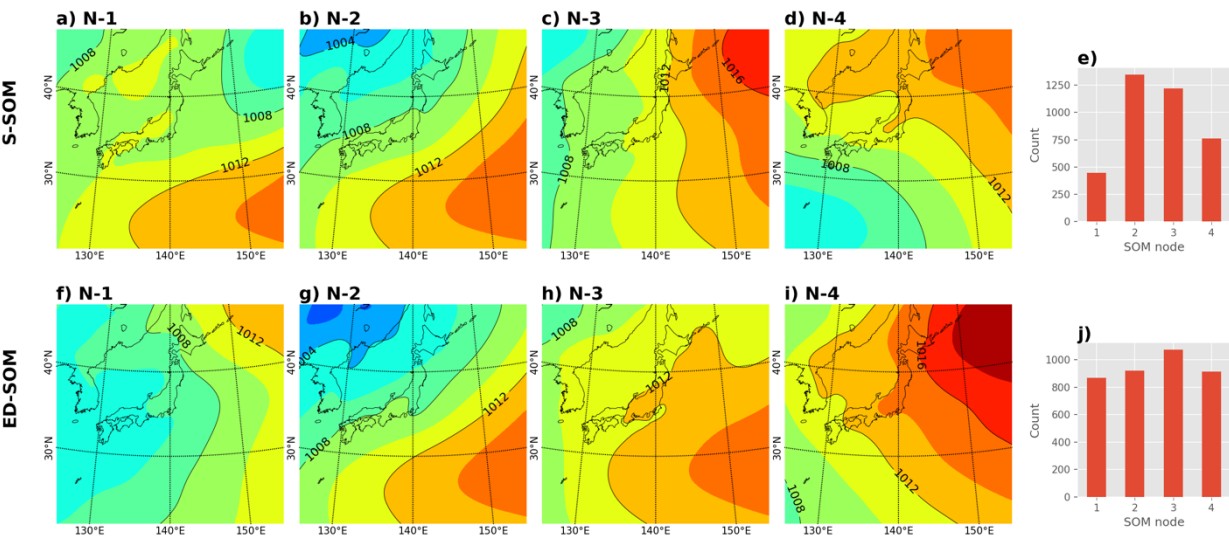

**Fig. A3. Summer (JJA) MSLP pattern revealed by SOMs. Upper plots, from a) to d), show S-SOM patterns (4 nodes); e) shows the number of daily MSLPs classified as nodes 1 to 4. The lower plots show the corresponding number for ED-SOM.**




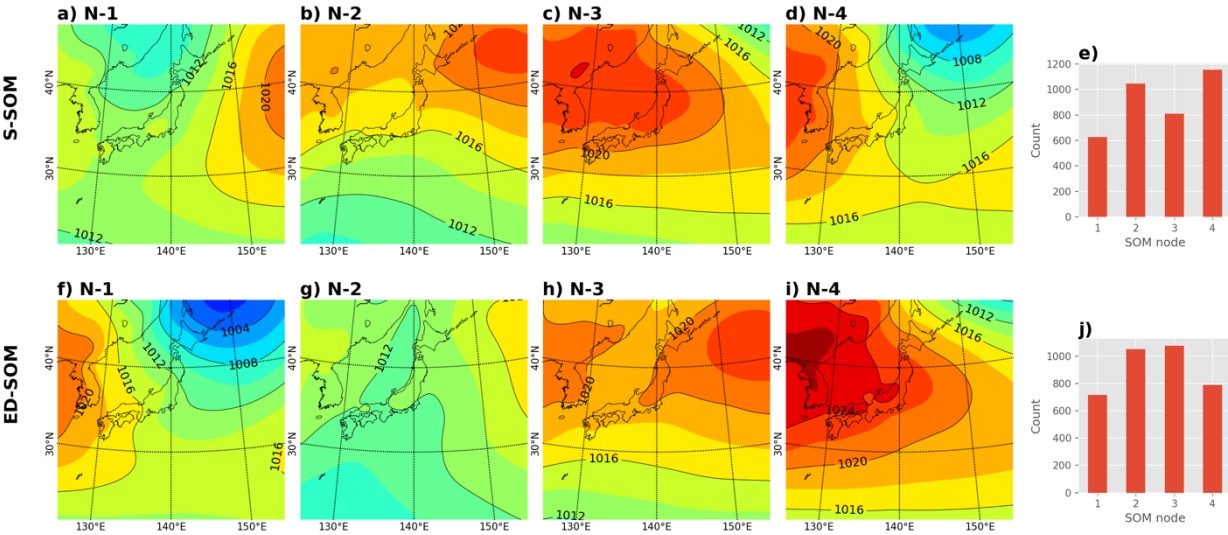

**Fig. A4. Autumn (SON) MSLP patterns revealed by SOMs. Upper plots, a) to d), show S-SOM patterns (with 4 nodes); e) shows the number of daily MSLPs classified as nodes 1 to 4. The lower plots show the corresponding results for ED-**
**SOM.**




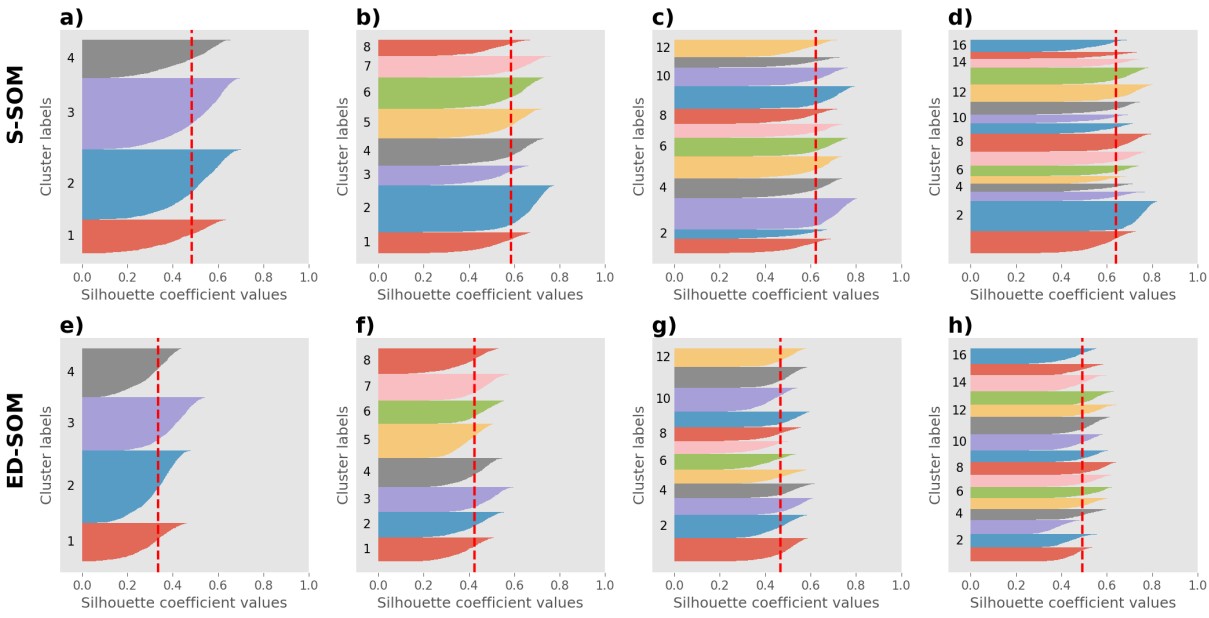

**Fig. A5. Silhouette plots for S-SOM and ED-SOM clustering for spring month MAM. a) to d) show results from simulations with 4, 8, 12, and 16 SOM nodes, respectively. e) to h) show the corresponding results for ED-SOM. The vertical red dashed lines in each plot indicate the silhouette score.**




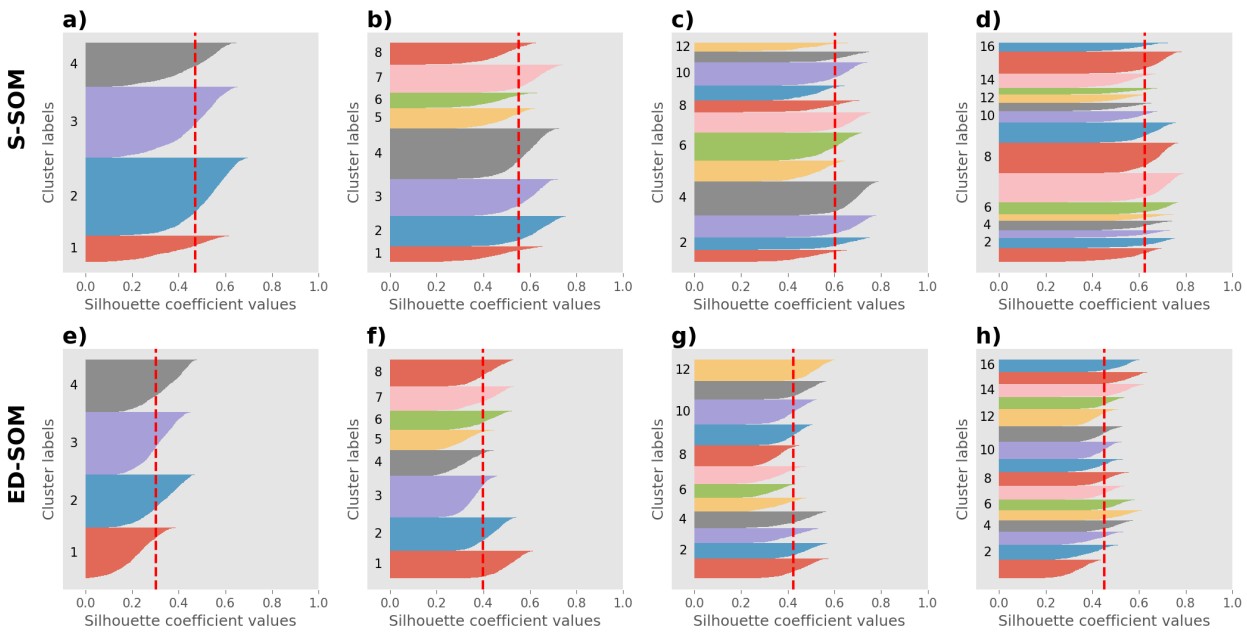

**Fig. A6. Silhouette plots for S-SOM and ED-SOM clustering for summer months JJA. a) to d) show results obtained**
**from simulations with 4, 8, 12, and 16 SOM nodes, respectively. e) to h) show the corresponding results for ED-SOM.**
**The vertical red dashed lines in each plot indicate the Silhouette score.**




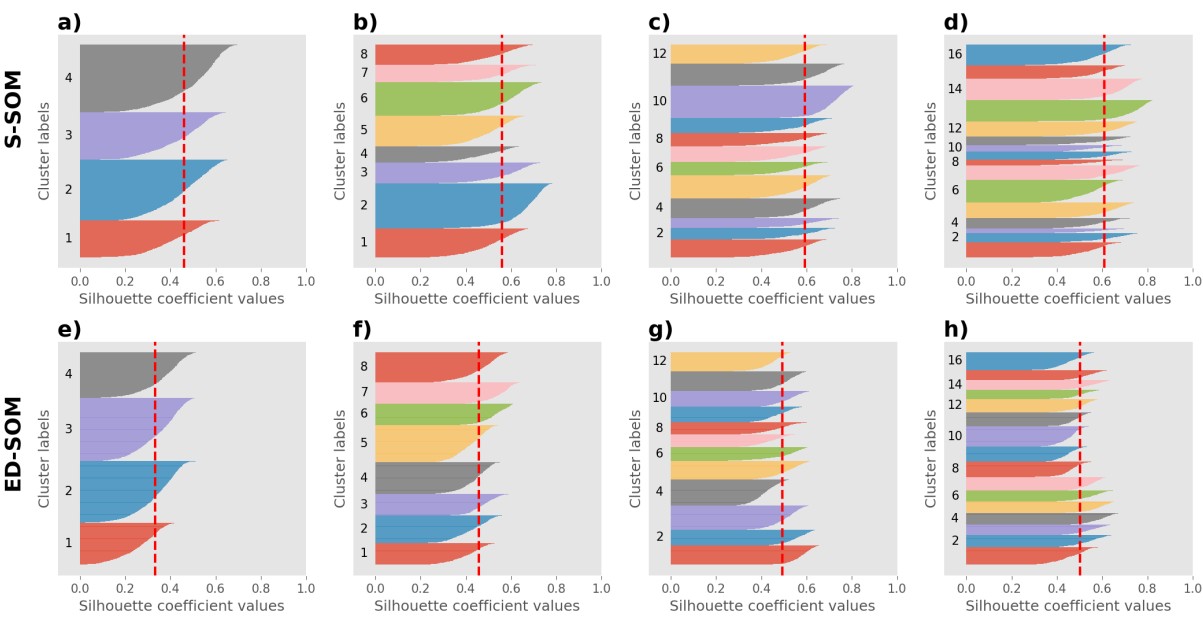

Fig. A7. Silhouette plots for S-SOM and ED-SOM clustering for autumn months SON. a) to d) show results obtained from simulations with 4, 8, 12, and 16 SOM nodes, respectively. e) to h) show the corresponding results for ED-SOM. The vertical red dashed lines in each plot indicate the Silhouette score.