# Peer review of "A structural self-organizing map (S-SOM) algorithm for weather typing"

_Geoscientific Model Development, 2020_

## Referee Comment (RC1) · Anonymous Referee #1 · 26 Oct 2020

Comments on "A structural self-organizing map (S-SOM) algorithm for weather typing" by Doan et al., submitted for GMD.

This manuscript is GMD-2020-27

Reviewer's suggestion: Minor revision

This study proposed a novel structural self-organizing map algorithm for synoptic weather typing. From the comparison to the traditional SOM using the Euclidean distance, the authors show the novel S-SOM method performance superior to a standard SOM with Euclidean distance. The results are interesting and the useful for the user of SOM in meteorological view. In addition, the manuscript is written and organized well. However, the manuscript needs some minor revisions before it can be considered for

publication, which can potentially contribute to enhance the value of the paper.

Specified comments

1. Isn't it possible to alleviate the problems that arise with ED-SOM by using correlation coefficients? That is included is as "structure" in S-SIM when c3=0. What is the use of advantage of S-SIM compared to the correlation coefficient? In addition, can we get better results than if we used the correlation coefficient?

2. L89: What do the three comparison measurements (luminance (ðÍŚŹ), contrast (ðÍŚŘ), and structure (ðÍŚă)) mean? How can we consider the use of each? For example, what should we do with the coefficients (such as c1 and ðÍŽij) if we want to change the SSIM to fit the purpose of the SOM's use?

3. L91: "$\sigma$"shoud be "$\sigma$x2". Please re-check the all formula used in the manuscript.

4. L91: Please include standard deviation "$\sigma$x".

5. How much will the calculation cost and time increase compared with ED-SOM?

6. Fig. 6/7: How about in a larger number of SOM nodes (such as 100/200/300/400)? Please include some additional information in the revised manuscript.

7. The authors use MSLP. Would the results be the same for other variables such as UV vectors or gradient of MSLP (difference from regional mean)?

8. In cases where the intensity of weather type plays a more important role than structure (e.g. prediction), it may be possible that ED may give better results?

---

## Referee Comment (RC2) · Anonymous Referee #2 · 28 Oct 2020

[referee-annotated manuscript omitted]

---

## Short Comment (SC1) · 14 Nov 2020

Dear authors,

in my role as Executive editor of GMD, I would like to bring to your attention our Editorial version 1.2:

https://www.geosci-model-dev.net/12/2215/2019/

This highlights some requirements of papers published in GMD, which is also available on the GMD website in the 'Manuscript Types' section:

http://www.geoscientific-model-development.net/submission/manuscript_types.html

In particular, please note that for your paper, the following requirement has not been

met in the Discussions paper:

- "The main paper must give the model name and version number (or other unique identifier) in the title."

Please add a version number for S-SOM in the title upon your revised submission to GMD.

Yours,

Astrid Kerkweg

―――――――――――――――――

---

## Author Comment (AC1) · 2 Jan 2021

Thank you. We have revised the title to: "S-SOM v1.0: A structural self-organizing map algorithm for weather typing"

---

## Author Comment (AC4) · 2 Jan 2021

Thank you. We have revised the title to: "S-SOM v1.0: A structural self-organizing map algorithm for weather typing"

---

## Author Response (AR1)

Astrid Kerkweg

a.kerkweg@fz-juelich.de

Dear authors,

in my role as Executive editor of GMD, I would like to bring to your attention our Editorial version 1.2:

https://www.geosci-model-dev.net/12/2215/2019/

This highlights some requirements of papers published in GMD, which is also available on the GMD website in the 'Manuscript Types' section:

http://www.geoscientific-model-development.net/submission/manuscript_types.html

In particular, please note that for your paper, the following requirement has not been met in the Discussions paper:

• "The main paper must give the model name and version number (or other unique identifier) in the title."

Please add a version number for S-SOM in the title upon your revised submission to GMD.

Yours,

Astrid Kerkweg

Thank you. We have revised the title to:

*"S-SOM v1.0: A structural self-organizing map algorithm for weather typing"*

**Response to Anonymous Referee #1**

**Reviewer's suggestion: Minor revision**

This study proposed a novel structural self-organizing map algorithm for synoptic weather typing. From the comparison to the traditional SOM using the Euclidean distance, the authors show the novel S-SOM method performance superior to a standard SOM with Euclidean distance. The results are interesting and the useful for the user of SOM in meteorological view. In addition, the manuscript is written and organized well. However, the manuscript needs some minor revisions before it can be considered for publication, which can potentially contribute to enhance the value of the paper.

We appreciate the reviewer for the constructive and helpful comments for improving the quality of this paper. We have carefully addressed all these comments, conducting additional tests and analysis suggested by the reviewer. The manuscript has also been revised accordingly.

Before going into further details, we would like to keep the reviewer notified of **one miscoding we found in line 73** of python script "*silhouette_anal.py*" (to calculate the silhouette scores), **but we confirmed that this miscoding does not largely affect the conclusions**. The original code was "*abis = np.argsort(ab)[::-1]*". The correct code must be "*abis = np.argsort(ab)*". This miscoding affected term $b$ in the silhouette-score formula: $s = (b - a)/(\max(a, b))$, where $b$ is the mean distance between a sample and all other points in the *next nearest cluster*. However, with original code, $b$ became the mean distance between a sample and all other points in the *farthest cluster*. Consequently, silhouette scores had been incorrectly overestimated. We have corrected the python script (https://zenodo.org/deposit/4437954#) and re-calculated all the silhouette scores. The correction has changed the score's absolute values, but it likely does not affect this study's conclusion. With the newly calculated silhouette score, S-SOM still exhibits the consistently higher performance over ED-SOM.

**Specified comments**

1. Isn't it possible to alleviate the problems that arise with ED-SOM by using correlation coefficients? That is included is as "structure" in S-SIM when c3=0. What is the use of advantage of S-SIM compared to the correlation coefficient? In addition, can we get better results than if we used the correlation coefficient?

We have examined the performance of SOM using the Pearson correlation coefficient (i.e., "structure" in S-SIM when $c_3$=0), hereafter called COR-SOM. As the reviewer speculated, the results show that COR-SOM performed better than ED-SOM (in terms of both the silhouette score and topographic error), particularly when cluster number is small (see Fig. R1, 2 below, or Fig. 6, 7 in the main text). Indeed, COR-SOM has comparable performance to S-SOM in experiments MAM and SON. However, COR-SOM is lower scored than S-SOM in DJF and partly in JJA.

We have revised the manuscript and figures accordingly adding the results of COR-SOM

[Figure]

**Fig. R1. Silhouette scores of S-, COR- and ED-SOM.** a) the winter experiment DJF; b) spring MAM; c) summer JJA, and d) autumn SON. In each plot, the x-axis denotes different SOM size configurations, and the y-axis represents the silhouette score that ranges from -1 to 1, with 1 means the perfect cluster assignment, 0 means a sample is located at the edge, a negative value means wrong cluster assignment.

[Figure]

**Fig. R2. Topographic errors of S-, COR- and ED-SOM.** a) the results for winter DJF; b) for spring MAM; c) for summer JJA, and d) for autumn SON. In each plot, the x-axis denotes different SOM size configurations, and the y-axis denotes topographic errors, with the minimum value of 1 indicating "no error" or best topographical preservation.

2. L89: What do the three comparison measurements (luminance (ðISZ), contrast (ðISR), and structure (ðISaˀ)) mean? How can we consider the use of each? For example, what should we do with the coefficients (such as c1 and ðIZij) if we want to change the SSIM to fit the purpose of the SOM's use?

- The S-SIM index was invented in image processing field primarily for measuring the similarity between two digital images. Three components of S-SIM, i.e., "luminance", "contrast", and "structure" represent human visual perception, i.e., "luminance" is to measure the similarity in brightness values; "contrast" is to quantify the similarity in illumination variability; and "structure" is to measure the correlation in spatial inter-dependencies between images (Wang and Bovik, IEEE Signal Processing Magazine, 2009, https://ieeexplore.ieee.org/document/4775883). We have added the additional explanation about the three comparison measurements into the revised text (Line 93 - 96).

- The damping coefficients $c_1, c_2$ are used to maintain the computational stability. Although, technically, we can consider adjusting $c_1, c_2$ to change the SSIM, we don't suggest tuning $c_1, c_2$ to fit the SOM purposes. We suggest

determining $c_1, c_2$ according to Wang and Bovik (IEEE Signal Processing Magazine, 2009) purely to avoid the computational instability.

3. L91: "σ"shoud be "σx2". Please re-check the all formula used in the manuscript.

We have revised "$\sigma$" to "$\sigma^2$" in the manuscript (Line 93).

4. L91: Please include standard deviation "σx".

We have included the standard deviation into the revised manuscript (Line 92).

5. How much will the calculation cost and time increase compared with ED-SOM?

The computational time of three SOM algorithms is shown in the following figure. S-SOM needs more computational cost, but it doesn't produce serious issue for the research. It is because total computational time is very small (less than 1 min) unlike the numerical weather prediction or climate change projection. For instance, ED-SOM needs 1 – 3 seconds to complete the jobs with node size ranging from 4 – 20. To do the same jobs, S-SOM needs 8 – 40 seconds, i.e., 10 – 15 times of ED-SOM; COR-SOM needs 8 – 40 seconds, i.e., 8 – 13 times of ED-SOM. We have added the figure below and text regarding the computational cost into the revised manuscript (Line 210 – 214).

[Figure]

**Fig. R3. Computational time of S-SOM, COR-SOM, and ED-SOM.** The x-axis indicates the SOM size configurations; the y-axis represents the elapsed time needed to complete the jobs. Note that this is the results from the DJF experiments with 3669 input sample, whose size is 65 x 72 pixels; the iteration steps of 5000 for each simulation.

6. Fig. 6/7: How about in a larger number of SOM nodes (such as 100/200/300/400)? Please include some additional information in the revised manuscript.

We have tested the SOMs a large number of nodes (e.g., 100/200/300/400) according the reviewer's suggestion. The results show the performance of SOM clustering getting lower with the increasing number of nodes. Look at the following figure for the silhouette score. At the size of 100/200/300/400, the score turns to be negative. Note that a silhouette score of below 0 indicates a sample has been assigned to a wrong cluster. We have added this information in the revised manuscript (Line 114 - 115).

[Figure]

**Fig. R4. Average silhouette scores of S-, COR- and ED-SOM.** The x-axis indicates the SOM size configuration with large ones at the right side. The y-axis indicates the silhouette score, which ranges from -1 to 1. For a give sample, the perfect cluster assignment has value of 1, negative values mean wrong cluster assignment for a sample.

7. The authors use MSLP. Would the results be the same for other variables such as UV vectors or gradient of MSLP (difference from regional mean)?

We have examined the SOM performance for wind vectors and gradient (difference from regional mean) of MSLP in accordance with the suggestion of the reviewer. To come to the point, both tests show the consistent results with the MSLP simulations, that is the superiority of S-SOM and COR-SOM versus ED-SOM in terms of both the silhouette score and topographical errors. Here we show few figures about the additional tests as following for references. We have added additional information about wind vector experiment into the revised manuscript (Line 205 - 209). Since we feel the "gradient of MSLP" is not different to "MSLP" in terms of data type to feed the SOM, we didn't add it into the manuscript.

[Figure]

**Fig. R5. Spatial patterns of the winter 500-hPa wind vector pattern revealed by SOMs with size of 4.** a) to c) show the S-SOM results ad d) shows the silhouette analysis plot; f) to j) shows the same results but for COR-SOM and k) to o) for ED-SOM. Input data are daily-base data (at 00 UTC) winter months DJF from 2009 – 2019.

[Figure]

[Figure]

**Fig. R6. Performance of S-, COR- and ED-SOM for winter 500 hPa wind vector clustering.** a) the silhouette scores; b) topographic errors of SOMs at different node size.

[Figure]

**Fig. R7. Spatial patterns of the winter gradient MSLP pattern revealed by SOMs with size of 4.** a) to c) show the S-SOM results ad d) shows the silhouette analysis plot; f) to j) shows the same results but for COR-SOM and k) to o) for ED-SOM. Input data are daily-base data (at 00 UTC) winter months DJF from 2009 – 2019.

[Figure]

**Fig. R8. Performance of S-, COR- and ED-SOM for winter gradient MSLP clustering.** a) the silhouette scores; b) topographic errors of SOMs at different node size.

8. In cases where the intensity of weather type plays a more important role than structure (e.g. prediction), it may be possible that ED may give better results?

We are sorry that we didn't get the meaning of "intensity of weather type". The primarily purpose is to provide the algorithm that can deal with spatial- (or temporal-) structured data. We demonstrated that S-SOM performs better vs ED-SOM for data such as 2-D MSLP and wind vectors. On the other hand, in the case, spatial or temporal structure do not matter (data are not map or time series), ED-SOM could be also useful.

**Response to Anonymous Referee #2**

The study proposed a novel method to capture the best matching unit in a self- organizing maps algorithm. The authors proved that this method is better suited for clustering meteorological data than a more widely used Euclidean distance method. This is an interesting and well written manuscript.

There are some minor housekeeping corrections / suggestions that the author can find in the attached pdf. Other than that the manuscript is acceptable to be published in my point of view.

Please also note the supplement to this comment: https://gmd.copernicus.org/preprints/gmd-2020-278/gmd-2020-278-RC2- supplement.pdf

We appreciate the reviewer for the constructive and helpful comments for improving the quality of this paper. We have carefully addressed all these comments, conducting additional tests and analysis suggested by the reviewer. The manuscript has also been revised accordingly.

Before going into further details, we would like to keep the reviewer notified of **one miscoding we found in line 73** of python script "*silhouette_anal.py*" (to calculate the silhouette scores), **but we confirmed that this miscoding does not largely affect the conclusions**. The original code was "*abis = np.argsort(ab)[::-1]*". The correct code must be "*abis = np.argsort(ab)*". This miscoding affected term $b$ in the silhouette-score formula: $s = (b - a)/(\max(a, b))$, where $b$ is the mean distance between a sample and all other points in the *next nearest cluster*. However, with original code, $b$ became the mean distance between a sample and all other points in the *farthest cluster*. Consequently, silhouette scores had been incorrectly overestimated. We have corrected the python script (https://zenodo.org/deposit/4437954#) and re-calculated all the silhouette scores. The correction has changed the score's absolute values, but it likely does not affect this study's conclusion. With the newly calculated silhouette score, S-SOM still exhibits the consistently higher performance over ED-SOM.

**Comments:**

Line 78 There are more than one method to find the BMU. Usually, Euclidean or Karl Pearson distances are used, but distance can be defined by any measure that might be appropriate to the particular problem. Here the authors introduce a new one and CHOSED one other popular method to compared to. It would be better to state that clearer since there are other methods that can deal with structural data, for example using correlation coefficient.

We have examined the performance of SOM using the Pearson correlation coefficient (i.e., "structure" in S-SIM when $c_3=0$), hereafter called COR-SOM. As the reviewer speculated, the results show that COR-SOM performed better than ED-SOM (in terms of both the silhouette score and topographic error), particularly when cluster number is small (see Fig. R1, 2 below, or Fig. 6, 7 in the main text). Indeed, COR-SOM has comparable performance to S-SOM in experiments MAM and SON. However, COR-SOM is lower scored than S-SOM in DJF and partly in JJA.

We have revised the manuscript and figures accordingly adding the results of COR-SOM.

[Figure]

**Fig. R1. Silhouette scores of S-, COR- and ED-SOM.** a) the winter experiment DJF; b) spring MAM; c) summer JJA, and d) autumn SON. In each plot, the x-axis denotes different SOM size configurations, and the y-axis represents the silhouette score that ranges from -1 to 1, with 1 means the perfect cluster assignment, 0 means a sample is located at the edge, a negative value means wrong cluster assignment.

[Figure]

**Fig. R2. Topographic errors of S-, COR- and ED-SOM.** a) the results for winter DJF; b) for spring MAM; c) for summer JJA, and d) for autumn SON. In each plot, the x-axis denotes different SOM size configurations, and the y-axis denotes topographic errors, with the minimum value of 1 indicating "no error" or best topographical preservation.